# An Intelligent Driver Training System Based on Real Cars

**DOI:** 10.3390/s19030630

**Published:** 2019-02-02

**Authors:** Gui-Jiang Duan, Xin Yan, Hong Ma

**Affiliations:** 1School of Mechanical Engineering and Automation, Beihang University, Beijing 100191, China; gjduan@buaa.edu.cn; 2Artificial Intelligence and Safe Driving Behavior Research Center, Beijing 100071, China; hblajx@126.com

**Keywords:** driver training, sensors, data acquisition, vocational skills education

## Abstract

In driver training, the correct observation of the trainees’ operation is the key to ensure the training quality. The operation of the vehicle can be expressed by the vehicle state changes. This paper proposes a driver training model based on a multiple-embedded-sensor net. Six vehicle state parameters are identified as the critical features of the reverse parking machine learning model and represented quantitatively. A multiple-embedded-sensor net-based system mounted on a real vehicle is developed to collect the actual data of the six critical features. The data collected at the same time are bound together and encapsulated into a vector and sequenced by time with a label given by the multiple-embedded-sensor net. All vectors are evaluated by subjective assessment conclusions from experienced driving instructors and the positive ones are used as the training data of the model. The trained model can remind the driver of the next correct operation during training, and can also analyze the improvements after the training. The model has achieved good results in practical application. The experiments prove the validity and reliability of the proposed driver training model.

## 1. Introduction

Car driving is a complex skill that relies on the multi-organ synergy of the hands, eyes, feet, and head, as well as real-time analysis and decision making of the brain. Driver training is a process in which driving knowledge, skills, and experiences are imparted to the trainees. Driver training is essential for assessing and providing drivers with adequate skills to drive in complex and dynamic environments [1]. Currently, most people still follow a coach to learn driving skills in driving schools. Due to the knowledge, experience, enthusiasm, expression ability, and cultural background limitations of human coaches, as well as the limitation of human communication means (verbal, body language, eye contact, etc.), complaints that training is inefficient and inadequate are frequent. In fact, there is a high probability of accidents among new drivers [2,3,4].

At present, the study of different aspects of driving training is very extensive [3,4,5,6,7,8,9,10,11], such as driving training for special people [7,12,13], accident prevention [6,14,15], driver reaction times [2,16], etc. However, we are not sure whether the training topics are effective or not [4,5]. In the training of novice drivers, driver training projects are mainly developed without a clear theoretical basis [17]. The definition of driving training used in this paper comes from the subjects in the driving skills identification test stipulated by the Chinese government. For clarity of expression, this paper calls these subjects the stipulated subject. In this paper, quickly and smoothly passing the stipulated subject is taken as the evaluation basis of training effect.

As technology has developed, computer-aided systems have been widely applied in the education or training fields [18,19,20,21], Bačić [22] and SAP Sports One group [23] used wearable sensors to collect human motion data and train in operational skills. Until now, a lot of driver feedback programs have been designed, each trying to cover as many aspects of driving as possible. Especially, Malik and Husnain [3,17] collected data from the driver, vehicle and environment, and designed a comprehensive Intelligent Drive Training System. Compared with previous studies, this paper will focus on helping the trainer without any driving experience to master the driving skills to pass the stipulated subject. In China, people who train in the stipulated subject often have no driving experience. Once the trainee begins to practice a real car, they would usually be fearful [24], this will affect their control of the vehicle [25]. Therefore, we have developed a driver training system mounted in the learner-driven vehicle. To the trainer, the real car-based system is advantageous in:Enabling the trainees to intuitively and accurately feel the influence of their actions on the state and trajectory of the vehicle, and quickly establish and modify their operational skills through interaction with the training system;Enabling the trainees to quickly adapt and eliminate the tension caused by vehicle movement.

However, motor vehicle driving is a complex multi-objective decision-making process [1], and with a high real-time component. In addition, the trainee’s maneuver in each subject must be immediately successful. If the trainee makes a mistake, he will fail the exam. There were many challenges that must be faced in applying computerd in the field of driver training:Identifying and defining the critical parameters that affect the training results, and building a multi-parameter complex decision-making model;Delivering teaching information to trainees in an efficient, accurate and timely manner;Effectiveness evaluation of sample data and screening;

To solve these problems, multiple methods are integrated in developing the system, including embedded intelligent sensing, high-precision positioning, digital 3D maps, and multimedia human-computer interaction, etc. Additionally, artificial intelligence (AI) is applied. An artificial neural network is used to build a real-time decision-making module. The system is therefore named Artificial Intelligence-based Driver Training System (AI-DTS). The detailed work performed while developing AI-DTS is introduced in the following sections of this paper.

## 2. The Framework and Key Technologies 

### 2.1. Analyzing and Modelling the Typical Training Course

Normally, driving schools use the five typical subjects shown in Figure 1 for driver training, according to which the authorities certify the students’ driving skills [26].

The above five subjects are all supported by the AI-IDTS. Among them, the “reverse parking” one is the most difficult subject as well as the most challenge part of AI-DTS, in which the driver is required to complete a series of complex operations in a narrow “T-shaped polygon” which represents a parking space. Thus, the subject “reverse parking” is chosen as the example in this paper to illustrate the development and application of the AI-DTS. The design principles of the other four modules are the same as the “reverse parking” module, so the details are not repeated in this paper.

To implement the AI-DTS, it is necessary to consider the following:How to build a model to describe the driver’s behavior and its impact on the vehicle?What are the key factors that affect the ability of motorists to perform these actions?How to evaluate the driving behavior and give feasible feedback?Does the artificially intelligent system have the ability to facilitate the trainee?

It has been confirmed by the driver training experts that, to accomplish successful reverse parking, it is important to control the vehicle’s trajectory. The trainee should master the ability to control the movement of the vehicle in accordance with the desired trajectory. In this paper, a machine learning model is built based on the motor vehicle driving knowledge and training experiences. Some key critical factors affecting the vehicle trajectory are taken as the input parameters of the machine learning model to be processed, so as to analyze the drivers’ operation and give real-time feedback to help them master the skills of reverse parking.

According to the discussions with experienced driving instructors, predicting the appropriate moment to rotate the steering wheel for successful reverse parking is quite a complex decision problem. It is found that there are three critical factors for successful reverse parking, namely, the initial position of the vehicle, the appropriate moment to rotate steering wheel, and the speed of the vehicle, respectively. In addition, there are quite complex correlations among these factors. Take reverse parking as an example. As shown in Figure 2, at the first critical point, an earlier and faster rotation of the steering wheel will cause the final position of the vehicle to move to the right, an earlier and faster rotation of the steering wheel will cause the final position of the vehicle to move to the left; at the second critical point, the time and speed of rotating the steering wheel will also have effects on the final position of the vehicle. Similarly, as shown in Figure 3, the initial parking position will also affect the vehicle’s trajectory, leading to the failure of the training task.

Among these correlated factors, the moment of rotating the steering wheel is the most critical one that drivers have to face. The appropriate moment to rotate the steering wheel is a result of the interaction of these multiple factors, which means that in reverse parking, the driver should predict the appropriate moment to rotate the steering wheel based on the comprehensive consideration of these factors including the position of the vehicle, the speed of the vehicle, the angle of the steering wheel and the attitude of the vehicle in the T-shape polygon, respectively. For this reason, the AI-DTS is typically a nonlinear mapping system of interrelated and multidimensional data. Traditional logic reasoning model may not suitable for such problems meanwhile the artificial neural network has certain advantages in this respect, so the artificial neural network was chosen as the machine learning model for AI-DTS.

### 2.2. The Framework of AI-DTS

As shown in Figure 4, driver training systems can give operational advice and evaluations in real time based on student actions. In the training process, the system collects the vehicle status information and the vehicle position information through the sensor network installed on the teaching vehicle. The data collected during the training process are analyzed by the artificial neural network and real-time feedback guidance is given to the trainees according to the results of the analysis to correct the trainees’ operation of the vehicle. The trainee interacts with the training system through a human-computer interactive interface based on 3D scene and speech dialogue.

Obviously, this is an interdisciplinary system involving multiple fields such as machine learning, embedded system, attitude measurement, ergonomics, motor vehicle driving, etc. The driver training system contains the decision-making functions of the training system.

AI-DTS is developed based on a real-time high-precision vehicle state acquisition system and real-time decision feedback system. To this end, we built a multi-sensor network containing data processing units. The multi-sensor network handles the collected data to the artificial neural network for analysis and then feeds back the results to the trainees. The application scenario of the AI-DTS is shown as Figure 5. 

As shown in Figure 5, the vehicle state data is derived from sensors installed on the learner-driven vehicle and OBD port of the vehicle. Since the data sources are dispersed, data integration and association is necessary. The data gathered from different sensors are bound together through an embedded system mounted on the vehicle. Considering a set of data that occurred on the same time reflects the state of the vehicle at that moment, the vehicle state data gathered from different channels are time serialized based on a uniform time-clock by the data acquisition system. Subsequently, a vehicle state matrix is constructed to quantify and indicate the trainee’s operation in a training session. Subsequently, the fully trained driver training system is used to guide the driver’s operation according to the real-time vehicle state data received during the process of training, so as to help the trainee master the driving skill better and faster.

Before the system was developed, we recruited 1200 volunteers of different characteristics (age, gender and height). The volunteers ranged in age from 20 to 40 and ranged in height from 160 to 185. The number of female and male volunteers was almost equal. The age, gender and height distribution of volunteers is shown in Figure 6 and Figure 7. The volunteers were divided into three groups randomly. Group A included 1000 volunteers, while the remaining 200 volunteers were shared equally in Group B and Group C. All the volunteers had no driving experience at all and the data collected and analyzed in the process of system development comes from the actual training of the volunteers.

The development of AI-DTS is carried out in five phases. First, a driving training model which is more suitable for artificial intelligence processing is built, in which six parameters are identified to be the critical features. Secondly, an integrated framework for realizing and operating the system is proposed. Thirdly, more than 1000 volunteers for practical training are recruited and a large amount of the training data are collected synchronously by a specially designed system. Subsequently, the artificial neural network is trained using the collected training data. Finally, 200 volunteer trainees are randomly selected to learn driving skills using this system to assess the effectiveness. The assessment result supports the practicality of the system.

### 2.3. The Key Technology of Developing AI-DTS

#### 2.3.1. The Critical Feature for AI-DTS 

The state of vehicle keeps changing with the operation of the driver, as well reflects the operation efficiency of the driver. Totally six vehicle state parameters are chosen as the critical features of the AI-DTS. The selection and the correlation of the six features are discussed as follows:

(1) Identifying the critical features

Three pairs of features are identified as the critical features for a success reverse parking, which are the position-related features, orientation-related features, and speed-related features, respectively:

• Position-related features

In a limited place, the position accuracy for the DGPS is consistently better than that of plain GPS [27]. In AI-DTS, the position of the vehicle was provided by DGPS. The DGPS is built for a specific field where the base station is located. Thus, the coordinate data of both the parking location and the position & orientation of the vehicle throughout the site gotten from the DGPS are all in the local coordinate system of the DGPS. Since the AI-DTS is designed for general purposes, it is necessary to ensure the AI-DTS to be able to quickly match the different DGPS systems of the particular campuses, so an AI-DTS coordinate system which is independent of any particular DGPS coordinate system is constructed based on the features of a standard parking location, as shown in Figure 8. During training of the AI-DTS model, the coordinate data obtained from the DGPS system are all automatically transformed into the AI-DTS coordinate system by the coordinate transformation matrix. Thus, it is ensured that the training and decision making of the AI-DTS model is independent of the specific site. Once the AI-DTS is fully trained and is being to be applied to the different campus, the AI-DTS is easy to match the campus by configuring the transform matrix of the local coordinate system and the AI-DTS coordinate system. 

• Orientation-related features

As for the critical parameter impacting the orientation of a moving vehicle, it is obvious that the angle of the steering wheel (swa) controls the orientation of the vehicle. The state of the steering wheel can be presented by its angle. In order to get the angle of the steering wheel, where the steering wheel of the actual training vehicle was measured and calibrated by an angle sensor. The initial position is set as zero as well as the left and right limitation is set as −600 and 600 respectively. 

In AI-DTS, the parameter “head angle (ha)” is used to indicate the orientation of the vehicle in the location. As shown in Figure 9, the head angle is defined as the counterclockwise angle between the forward and the X-axis.

The goals of the driver training are teach trainees driving skills and help them pass the qualification examination. In reverse parking, the drivers should carefully maintain the appropriate trajectory of the vehicle and park the vehicle in the specified position. To do this, the driver should adjust the vehicle trajectory twice by rotating the steering wheel at two critical points. At the first critical point, the steering wheel should be rotated from the initial position to the limit position, while in the second critical point, the steering wheel should be rotated back (Figure 10). Rotating the steering wheel on a wrong point may lead to the shift in the final position of the vehicle parking. In training, it is important to help the trainee master the skills to determine the appropriate moment of adjusting the steering wheel according to the speed of the vehicle.

• Speed-related features

According to the principle of kinematics, the driving trajectory is determined by both the orientation and speed of the vehicle. Generally, the accelerator pedal determines the vehicle’s speed. However, for the beginners in the training course of reverse parking, the case is not quite like that. Experienced instructors usually find that, for beginners, the parking results are strongly related with the ability to control the vehicle speed and orientation, so in training, the instructors usually train the trainee in a gradual way from easy to difficult. For example, in the beginning stage of reverse parking training, the instructors usually require the trainees to maintain a lower speed so that they have enough time to balance the speed and orientation. For this reason, in a typical training process of reverse parking in China, the trainees are usually required to drive at idle speed. In this condition, the vehicle speed is not controlled by the accelerator pedal, but the clutch pedal. When the clutch pedal is depressed, the speed reduces, on the contrary, it speeds up, so, in the AI-DTS, the depth of the clutch pedal (dcp) is chosen as the assessment parameter of speed. 

In summary, the coordinate of the vehicle in the training location (x, y), the steering wheel angle (swa), the head angle (ha), the vehicle speed (vs) and the depth of the clutch pedals (dcp) are chosen as critical features of the AI-DTS and are encapsulated into a 6-dimension vector to be used as input to the AI-DTS. The assessment conclusions about the driver’s operation given by the experienced instructors are used as the target output for training the AI-DTS. Both of the input data and target output data mentioned above should be collected before the AI-DTS model is to be trained.

(2) Brief Discussion of the correlation between the six critical features 

In order to verify the validity of the selected six critical features, a small amount of training sample data is collected to analyze the correlation between the six critical features and the correlation between them and the trajectory.

Figure 11 presents a group of curves of the four features in a specific reverse parking operation, such as the swa-time curve (Figure 11a), the vs-time curve (Figure 11b), the ha-time curve(Figure 11c) and the dcp-time curve(Figure 11d), respectively.

As shown in Figure 9, initially, the swa keeps at about “0” and maintains a certain period of time. When runs to the first critical point, it declines from “0” to the value of about “−600” sharply and keeps a certain period of time also. Correspondingly, the ha curve stays at the initial value of about “−40” for a certain time. After the first critical point, the curve changes smoothly to the value about “−140”. Subsequently, at the second critical point, the swa curve restores to the value of approximately “0” sharply and keeps gradually stabilized after that. Similarly, the ha curve keeps in “−140” after the steering wheel resets. 

The above two curves reveal a significant synchronization between the steering wheel angle and the vehicle orientation. Similarly, the significant synchronization between the vehicle speed and the depth of the clutch pedals is also supported by the vs. curve (Figure 11c) and the dcp curve (Figure 11d). The corresponding trajectory of the vehicle is shown as Figure 11e. The synchronization between the vehicle trajectory and the other four features is also significant. This synchronization proves the validity of the selected six critical features.

#### 2.3.2. Model Selection for Machine Learning Modular

In the process of reverse parking, the appropriate moment to rotate the steering wheel has a significant impact on the trajectory. During the process of successful reverse parking, the reasonable angle of the steering wheel is dynamically related and interacted with a series of factors on that moment, including the relative position, relative orientation, speed and angle of the clutch pedal. As discussed in Section 1, the neural network is suitable for building a decision model to determine the appropriate moment to rotate the steering wheel based on the comprehensive consideration of the above factors. Also, the accuracy and efficiency of the decision model are important to AI-DTS. It is necessary to choose an appropriate active function to get reasonable accuracy and efficiency. The ReLU is the most popular activation function for deep neural networks in 2018 [28]. Compared with other functions, he has more advantages in training deep neural networks [29]. 

However, in the actual training, a lot of negative values are usually produced. In order to avoid losing the negative values and to promote the model training efficiency, LeakyReLU was chosen as the active function of the model. Logistic regression is used as the classifier of the artificial neural network. 

#### 2.3.3. Sample Data Preparation for Training the AI-DTS 

The sample data for training the AI-DTS are all derived from the actual data generated by actual motor vehicle driver training. For the safety of the experimenter, a closed-loop auto brake system was installed on the training vehicle. The brake system could automatically brake in advance when the vehicle is in danger.

The data collection system is mounted on a real vehicle. This is an integrated system consists of multiple sensors, signal receivers, signal processors and signal transmitters, respectively. In the process of driver training, the data collection system monitors and gathers the state data of the multiple parts such as the steering wheel, the clutch pedal, the parking brake, gears, the engine speed, the turn signal, the handbrake and the horn as well as the state data of the whole vehicle such as the position, the speed and the attitude, through the sensor network in a frequency of 10 Hz. All the above state data at the same time are bound together and encapsulated into a dataset called “vehicle state dataset”. The participants are chosen randomly from the trainees in a driver training institution. All of them are volunteers to the study. They have been informed which data the study will collect and the purpose of the study. The training is not disturbed by the data acquisition. 

In the following paragraphs, three critical aspects for preparing the sample data of the AI-DTS are presented.

(1) Sample data preparation for training the AI-DTS 

To get the sample data for training the AI-DTS, both the vehicle state data (as the input data) and the instructor’s assessment conclusion (as the target output data) have to be collected and associated. As for the former, the instantaneous state of the vehicle is recorded in the frequency of 10 Hz by the sensor network installed on the learner-driven vehicle. Meanwhile, the assessment conclusion of the rationality of the driver’s operation given by the experienced driver instructors should be firstly quantified and then bound with the corresponding vehicle state data set. To facilitate the manipulation of sample data collecting, a software tool which is integrated with the sensor network is developed and named as AI-DTS-Sampler. The user interface of the AI-DTS-Sampler is shown as Figure 12. When sampling the driver training data, the software tool is operated by the experiment engineer. Before sampling a particular reverse parking training, the basic information of this case such as the training position, the training subject is input through the corresponding edit box. Once the reverse parking beginning starts, the sampling is launched. The vehicle state data is scroll-displayed in the edit box at the bottom of the window and saved into a particular data file. When the reverse parking operation is over, the assessment conclusion about the rationality of the driver’s operation given by experienced instructor is quantified via a slider bar and then saved into the corresponding data file to tag the matrix, “0” for appropriate, “1” for non- appropriate, respectively. Each training sample case is given a unique ID. 

The AI-DTS-Sampler is written in C++, but it can also be written in other programming languages. The algorithm of it can be applied in other similar fields to complete the data acquisition and preprocessing.

(2) The preprocessing of the raw sample data

As discussed in Section 2.3.1, a lot of training sample data for training the AI-DTS are collected from the actual training practices through the AI-DTS-Sampler. Whereas, before being used to train the AI-DTS, the raw sample data has to be specially preprocessed, removing the unqualified sample data and transforming to be the available format for the AI-DTS. The preprocessing consists of the following three steps:

• Time serialization of the sample data 

As mentioned in Section 2.3.1, six state parameters are chosen as the critical features of the AI-DTS, namely the coordinate of the vehicle in the training location (x, y), the angle of the steering wheel (swa), the head angle (ha), the vehicle speed (vs) and the depth of the clutch pedals(dcp), respectively.

During the sampling process, the six parameters are collected by the AI-DTS-Sampler in a frequency of 10 Hz. The values of the six parameters gathered at the same time point (for example *t*1) are first encapsulated into a 6-dimensional vector:(1)$$\left(x_{t1},y_{t1},swa_{t1},ha_{t1},vs_{t1},dcp_{t1}\right)$$

For a reverse parking operation, multiple such vectors can be collected in the frequency of 10 Hz. Subsequently, the multiple vectors of a specific reverse parking operation are time serialized to be assembled as a matrix of m × 6: (2)$$\left[\begin{array}{c} \begin{array}{cccccc} x_{t1}\; & y_{t1}\; & swa_{t1}\; & ha_{t1}\; & vs_{t1}\; & dcp_{t1}\; \end{array}\\ \begin{array}{cccccc} x_{t2}\; & y_{t2}\; & swa_{t2}\; & ha_{t2}\; & vs_{t2}\; & dcp_{t2}\; \end{array}\\ \vdots \\ \begin{array}{cccccc} x_{tm} & y_{tm} & swa_{tm} & ha_{tm} & vs_{tm} & dcp_{tm} \end{array} \end{array}\right]$$

The above matrix records the changing process of the vehicle state aroused by the driver‘s operation in reverse parking. Finally, the matrix is saved as a data file in txt format. The data file is named with the ID of the reverse parking operation given by the AI-DTS-Sampler.

• Sample data cleaning and assembling

Sample data collection covers a large number of reverse parking operation examples, including both successful ones and failed ones. Only the samples of the successful ones are useful for training the AI-DTS. The assessment conclusion about the rationality of the driver’s operation is used to clean the raw sample data. When the reverse parking operation is over, the experienced instructor gives an assessment conclusion about the rationality of the driver’s operation. The sample data matrix is tagged with the conclusion, “0” for rational, “1” for non-rational. Through data cleaning, only the sample data matrices tagged with “0” are selected. Then, the selected sample data was divided into two parts through the change of swa, the 6-dimensional vectors in the selected sample data with a degressive swa were picked and linked to be a matrix with the shape of “n×6” for the model to predict the right moment of rotating the steering wheel in the first critical point (AI-DTS-M1):(3)$$\left[\begin{array}{c} \begin{array}{cccccc} x_{t11}\; & y_{t11}\; & swa_{t11}\; & ha_{t11}\; & vs_{t11}\; & dcp_{t11}\;\\ x_{t21}\; & y_{t21}\; & swa_{t21}\; & ha_{t21}\; & vs_{t21}\; & dcp_{t21}\; \end{array}\\ \vdots \\ \begin{array}{cccccc} x_{tm1} & y_{tm1} & swa_{tm1} & ha_{tm1} & vs_{tm1} & dcp_{tm1}\\ x_{t12} & y_{t12} & swa_{t12} & ha_{t12} & vs_{t12} & dcp_{t12}\\ x_{t22} & y_{t22} & swa_{t22} & ha_{t22} & vs_{t22} & dcp_{t22} \end{array}\\ \vdots \\ \begin{array}{cccccc} x_{tm2} & y_{tm2} & swa_{tm2} & ha_{tm2} & vs_{tm2} & dcp_{tm2} \end{array}\\ \vdots \\ \begin{array}{cccccc} x_{tm3} & y_{tm3} & swa_{tm3} & ha_{tm3} & vs_{tm3} & dcp_{tm3} \end{array}\\ \vdots \\ \begin{array}{cccccc} x_{tmk} & y_{tmk} & swa_{tmk} & ha_{tmk} & vs_{tmk} & dcp_{tmk} \end{array} \end{array}\right]$$

Use “*k*” to express the times of the practice, Use “*n*” to express the sum of rows of all selected matrices. Then:(4)$$n=\sum_{i=1}^{k}m_{i}$$

Subsequently, all the other matrices are processed in the same way for the second critical point (AI-DTS-M1).

• Binarization of the steering wheel states

In the above matrix, the parameter swa, which represents the angle of the steering wheel is recorded as a float number ranges from −600 to 600, whereas, to perform a success reverse parking, the driver has to quickly rotate the steering wheel at the two critical points. As shown in Figure 11a, at the first critical point, the curve of swa declines sharply from 0 degrees to −600 degrees in 5 seconds. Similarly, at the second critical point, the curve of swa rises sharply from −600 degree to 0 degrees. That means, it is reasonable to binarize the steering wheel states parameter swa in two values, “0” for “non-rotated” and “1” for “rotated”. Considering the tolerance of the steering wheel angle sensor, the algorithm of binarization is described as shown in Figure 13: if the absolute change of swa is less than 100 degree, the steering wheel is inferred as “non-rotated” and then binarize the value of swa to “0”; on the contrary, if the absolute change of swa is over 100, the steering wheel is inferred as “rotated” and the value of swa is binarized as “1”. Thus, the values of the swa column of the above matrix are binarized as a series of “0” or “1” to present the states of the steering wheel. Subsequently, the swa column of the above matrix is taken out from the mother matrix and save as a matrix of n×1. This matrix is used as the output value for training the AI-DTS. In turn, the remaining five columns of the matrix are saved as a new matrix of n×5. This new matrix is used as the input value for training the AI-DTS.

#### 2.3.4. The HMI of AI-DTS

The visualization of the drive is an integral part of providing feedback to the driver [3]. The people who use AI-DTS are those who have not had enough driving experience. It is necessary to make the trainees intuitively feel the influence of his operation on vehicle trajectory change through human-computer interaction. The human-computer interface of the system provides three views, the top view, the rear view and the front view. Top view can help students intuitively understand the relative position of the current vehicle and the location, and help them to see the impact of their operations on the vehicle trajectory; The rear view can make trainers clearly understand the relative position relationship between the rear wheel and the warehouse line; the view angle of the front view is the same as that of the trainees in the vehicle, so that the trainees can better adapt to the interactive system, as shown in Figure 14.

In order to guide the trainees better and for the security of the trainees, we introduce human-computer voice interaction at the same time of human-computer interface and broadcast the adjust operation to the trainees through voice. Including prompt the start time, prompt rotate the steering wheel, prompting the operation error, etc. The steps of the voice prompts during the process of reverse parking are shown in Figure 15.

## 3. Performance Assessment

In this section, the test results of a well-trained AI-DTS system will be discussed so as to validate the usefulness of this model in practical training. The AI-DTS contains two machine learning models, the AI-DTS-M1 and the AI-DTS-M2, that are used to predict the first and the second critical points, respectively. 

As shown in Figure 16, at the beginning stage of reverse parking, the AI-DTS-M1 is first called by the AI-DTS for predicting the first critical point to provide operational guidance to the driver or assessing the rationality of the driver’s operation. Once the driver has rotated the steering wheel to the left-limit or right-limit at the first critical point, the AI-DTS-M2 will be activated instead of the AI-DTS-M1 to predicting the second critical point or assess the rationality of the driver’s operation for the second critical point. 

Before being applied in practical training, the AI-DTS needs a full test to validate the performance and function. The testing method includes two steps. First, the precision and the recall of the model were calculated in 9857 test samples (4041 for AI-DTS-M1 and 5816 for AI-DTS-M2). Second, the AI-DTS was used in the real reverse parking training by an experienced trainer, the trainer does reverse parking training 300 times, the subjective feedback from the trainer was recorded each time. Finally, we found 200 volunteers without any driving experience and divided them into two groups. One hundred volunteers learned driving skills using the Intelligent Driving Training System, and one hundred used traditional driving skills training methods. We compared their learning efficiency and the pass rate of the driver qualification examination. The test process and results are introduced in the following sections.

### 3.1. Performance Assessment Based on Statistical Methods 

The recall and precision are used as the indicators to assess the performance of AI-DTS in determining the critical points. As shown in Table 1, while predicting the appropriate moment of rotating the steering wheel in the first critical point, the precision of rotating the steering wheel is 0.95, the precision of non-rotating the steering wheel is 0.85, the recall of rotating the steering wheel is 0.93, and the recall of non-rotating the steering wheel is 0.88. While predicting the appropriate moment of rotating the steering wheel in the second critical point, the precision of rotating the steering wheel is 0.94, the precision of non-rotating the steering wheel is 0.93, the recall of rotating the steering wheel is 0.94, and the recall of non-rotating the steering wheel is 0.94. For driver training, the above indicators are all at an acceptable level.

For the performance of AI-DTS in determining the first critical point, the vehicle position is used to show the decision result. As shown in Figure 17, in the process of judging the first critical point to rotate steering wheel, the wrong judgments are evenly distributed.

### 3.2. Applicability Assessment Based on the Experimenter’s Intuitive Feedback

The AI-DTS was validated by the experiment in the real training field with the help of an experienced driver trainer. It was validated in three ways:The vehicle was operated by the experienced driver trainer, and the moment to rotate the steering wheel was predicted by the driver and the AI-DTS at the same time. In the 100 operations, there were 84 times in the first critical point and 91 times in the second critical point that the prediction made by the AI-DTS were in line with the driver’s operation;The vehicle was operated by the experienced driver trainer, and the appropriate time to rotate the steering wheel was judged by the driver and the AI-DTS at the same time. Stopped the vehicle in the critical point and observed the output of AI-DTS at the same time. In the 100 operations, there were 92 times in the first critical point and 87 times in the second critical point that the prediction made by the AI-DTS were changed while the vehicle was stopped;The appropriate moment to rotate the steering wheel was solo judged by the AI-DTS. The vehicle was driven by the driver according to the output of AI-DTS. Observed the parking position after parking. In the 100 operations, there were 94 times that the result was acceptable.

The results of the experimenter’s intuitive feedback are listed in Table 2.

### 3.3. AI-DTS’s Performance in Training Practice

Before introducing the experimental results, we must emphasize that the method of developing AI-DTS introduced in this paper is a general method. We can get models for the remaining four subjects by changing the critical points. The critical points of the rotating steering wheel in the remaining four subjects are shown in the Figure 18. The AI-DTS for the execution of training covers all subjects. In this section, all the results were obtained after the trainees completed all the training subjects under the guidance of AI-DTS.

After the volunteers completed the training, we analyzed the learning efficiency of the volunteers and pass rate of the driver qualification examination. Compared with the traditional teaching method, the learning efficiency of the intelligent driving training system is improved by 20%, and the pass rate of the driver qualification examination is increased by 22%. The experimental results are shown in Table 3.

Through the experiments, the decisions made by AI-DTS are similar to the decisions made by the experienced trainers. The test results indicate that the AI-DTS is practical for the real-life driver training. While the trainee drives the vehicle according to the advice given by AI-DTS, the parking position of the motor vehicle could satisfy the training requirements. Compared with traditional training methods, trainees using intelligent driver training system have higher learning efficiency and a higher pass rate of the driver qualification examination.

## 4. Discussion 

In the development of the model, we focus on the conventional operation scenarios. In conventional scenarios, vehicle status is used to define the operation of motor vehicle drivers in the training process and summarizes the operation that motor vehicle drivers need to complete the training in the backing-in training project. AI-DTS is used to identify the key position of two rotating steering wheels in the process of vehicle driver training, and the expected accuracy is achieved. The model results show that there is not a relatively fixed position of the rotating steering wheel during the training process for the driver of the motor vehicle. The position of the vehicle on the training ground, the speed of the vehicle and the angle between the vehicle and the garage will affect the key position of the rotating steering wheel. In the process of AI-DTS training, we neglect the extreme situation. The training data we use is the vehicle status data during the driver training. In the face of some edge data which will cause the vehicle to take any operation cannot complete the training, AI-DTS will not stop the calculation, but will give the key position of the current vehicle status. This is clearly an unreasonable treatment. In our future work, we will solve the extreme cases by optimizing the model.

Once rotated the steering wheel in the wrong position, it is hard to remedy the mistake and complete a correct reverse parking operation by adjusting the other parameter. Controlling vehicle trajectories to adapt to the different rotating position is considered an even more difficult accomplishment. In spite of many good trainers and good drivers work for passing the driver qualification examination, it is difficult to pass the examination if the driver rotating the steering wheel at a wrong position.

Therefore, from the author’s point of view, it is suggested that the beginners should first focus on the control of the vehicle, rather than looking for the critical point of the rotating steering wheel in the driver training. In the initial stage of training, the coach can help beginners to master the basic skills to operate or control the vehicle, such as changing the vehicle’s movement direction, keeping the vehicle at a very low speed forward, rotating the steering wheel and so on. At the beginning stage of the training course, if it is focused on mastering the timing to rotate the steering wheel, the trainees tend to overlook basic operational training. Once some wrong basic operating habits are developed, it is difficult to correct the training in the back to correct it later.

In order to reduce the human error in gathering, combining and serializing the vehicle status data and trajectory data in real time, a set of data acquisition software was specially developed. This software is developed in C++. It can serialize and store the data in the time of receiving the data from the sensor network mounted on the real vehicle. In the training process, the software will remind the coach following the completion of the training to evaluate the operation of the driver in time. Then it would label the collected data no matter the data is evaluated or not. For the scientific community, if the data was accepted by the sensor network and need to be evaluated periodically the similar solutions work. For example, it can be used to collect, to combine and to serialize the data of the parameter of the patient’s body and the doctor’s diagnosis.

Vehicle manufacturers may integrate all vehicle status parameters into the OBD system in the future, which will make it more accurate to obtain vehicle status parameters. The development of intelligent driving aids, vehicle networking technology, and big data technology makes this situation more and more possible.

## 5. Conclusions

This paper has discussed the development and application of AI-DTS, a driver training system designed for providing real-time mentoring or post-assessment feedback in driver training, for example, predicting the appropriate moment to rotate the steering wheel in reverse parking. The cross-discipline contribution of this study lies in the following aspects.

Firstly, the AI-DTS provides sufficient evidence to introduce artificial intelligence for decision-making in the vehicle training. Decision-making model for driver training is a complex decision model influenced and constrained by multiple factors. A well-trained AI-DTS is able to predict the appropriate moment to rotate the steering wheel in real-time and in a reasonable accuracy (see Table 1). The subjective feedback from the experimenter (both of the instructor and the trainee) also confirms the contribution of the AI-DTS to driver training.

Secondly, the AI-DTS presents both an effective method and practical evidence for analyzing the human driving behavior qualitatively. Based on identifying and quantifying the six critical features, a quantitative model is constructed in the AI-DTS to describe the driver’s operation behavior and its impact on the vehicle. Further, regards the needs of machine learning, the methods to preprocess the unformatted sample data is clarified. 

Considering practicality, there are still some issues worth further discussion. For example, the appropriate depth of the clutch pedal for keeping half-linkage varies in different vehicles. It means that the AI-DTS used for a particular vehicle can only be trained using the sample data collected from the same vehicle. The AI-DTS for the different vehicle should be trained individually. Obviously, this will result in heavy duty for training sample collection. Likewise, individual differences in the vehicle’s steering wheel, idle speed, etc. can lead to the same problem, though not as noticeable as the clutch pedals. In addition, when the AI-DTS is used in different types of vehicles, the influence of vehicle dimensions, wheelbase, and other parameters should not be ignored. To improve the universality and practicability of the AI-DTS in future research, consideration can be given to introducing some adjustment factors into the AI-DTS model. When the AI-DTS is initialized for different vehicles or models, these adjustment factors can be quickly matched. Probably, the model or algorithm for identifying and matching these adjustment factors are also issues worth a study. Finally, the gender and age of volunteers are not distinguished in the comparative experiment. Perhaps the gender and age of volunteers will have an impact on the application of the driving training system. Further research is needed.

## Figures and Tables

**Figure 1 sensors-19-00630-f001:**
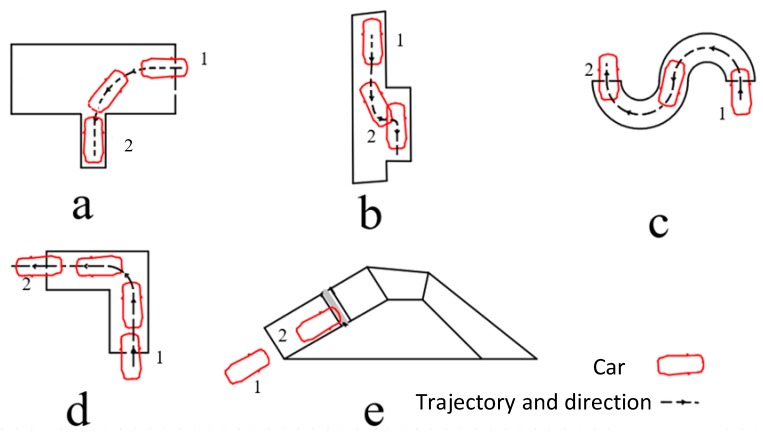
The five subjects of motor vehicle driver training course and driving license test in China: (**a**) Reverse parking, vehicles reverse from point 1 to point 2 in a T-shape polygon; and (**b**) parallel parking, vehicles reverse from point 1 to point 2 parallel; and (**c**) tortuous route, vehicles move from point 1 to point 2 following the “S” curve; and (**d**) quarter turn, vehicles move from point 1 to point 2 following a quarter turning and (**e**) ramp parking, vehicles move from point 1, parking in a set position on the road.

**Figure 2 sensors-19-00630-f002:**
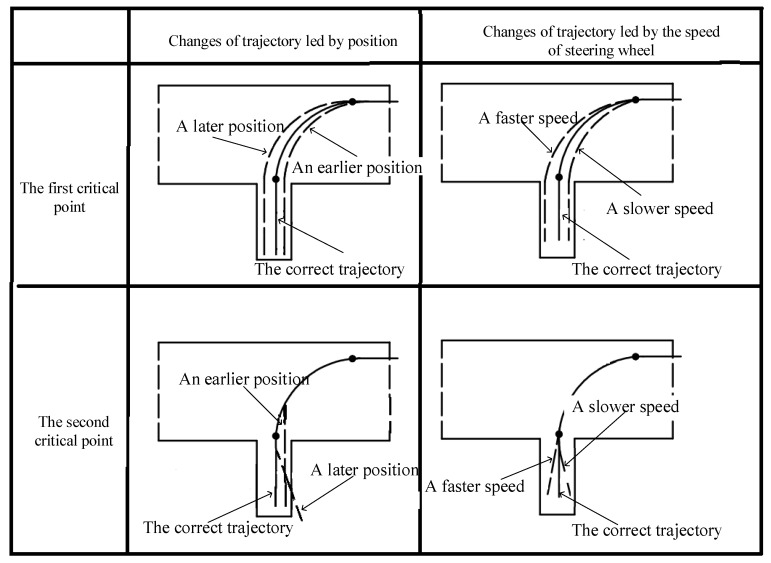
Two critical points affecting trajectories of reverse parking.

**Figure 3 sensors-19-00630-f003:**
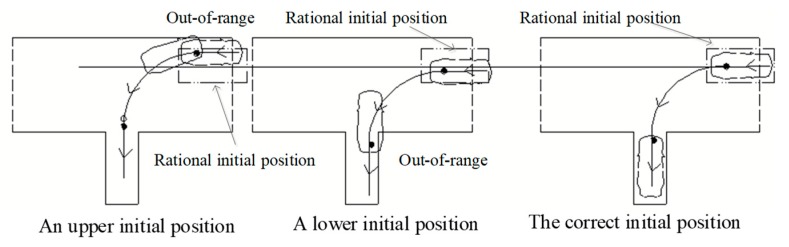
The initial position affecting trajectories of reverse parking.

**Figure 4 sensors-19-00630-f004:**
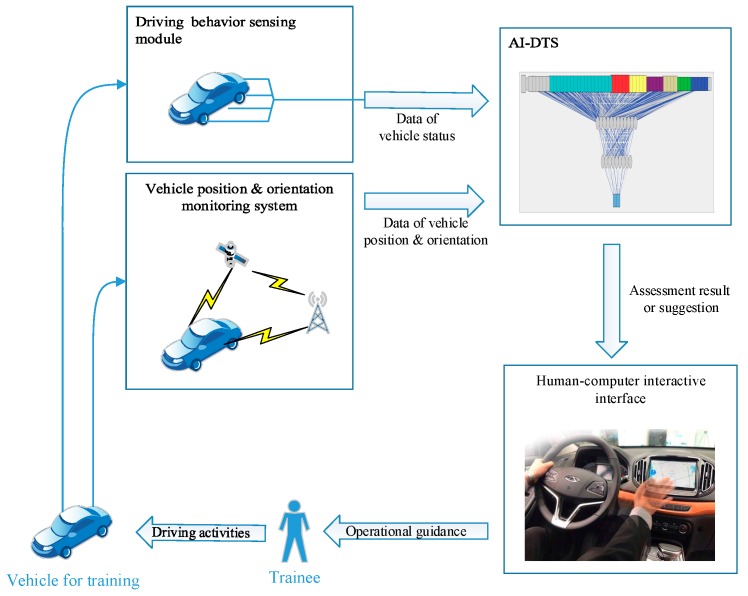
The framework and process model of AI-DTS.

**Figure 5 sensors-19-00630-f005:**
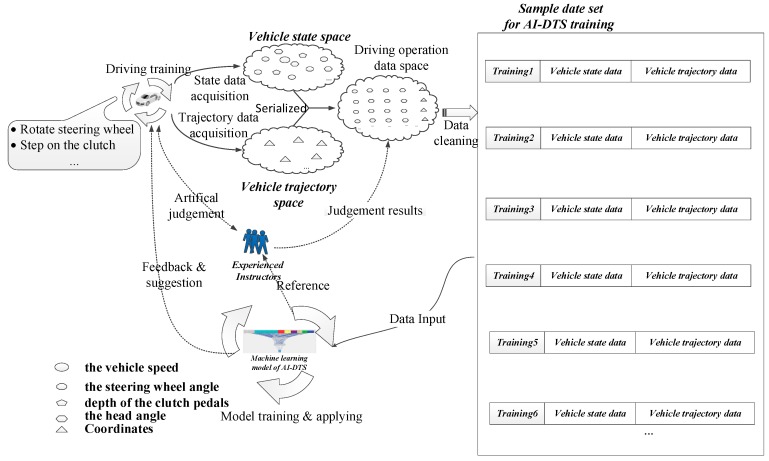
The framework and practice scenario of the AI-DTS.

**Figure 6 sensors-19-00630-f006:**
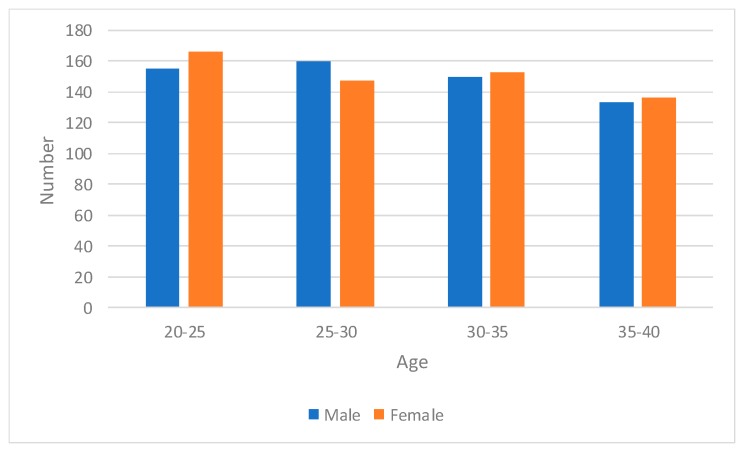
The age and gender distribution of volunteers.

**Figure 7 sensors-19-00630-f007:**
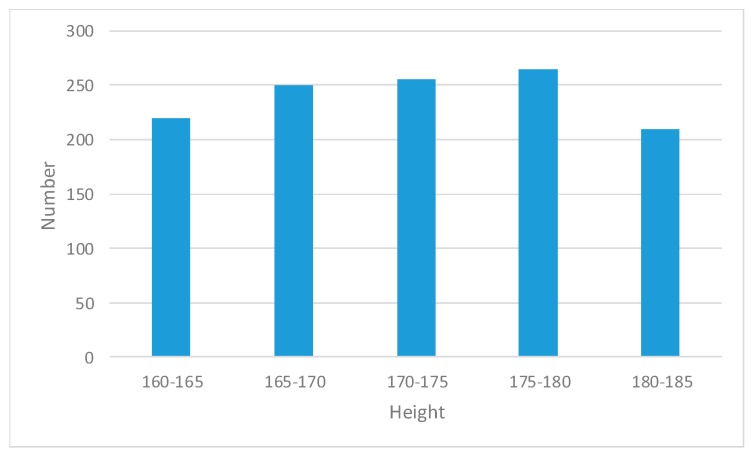
The height distribution of volunteers.

**Figure 8 sensors-19-00630-f008:**
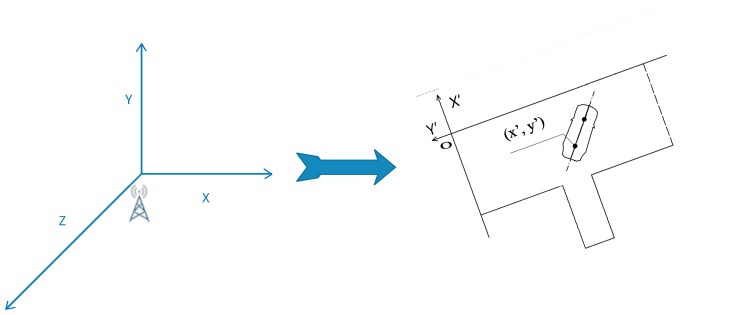
Coordinate transformation for the general purpose of AI-DTS.

**Figure 9 sensors-19-00630-f009:**
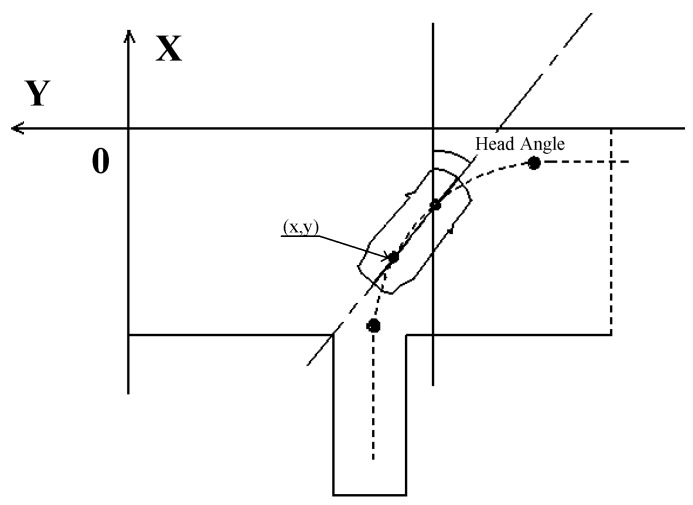
The head angle of the vehicle in the reverse parking location.

**Figure 10 sensors-19-00630-f010:**
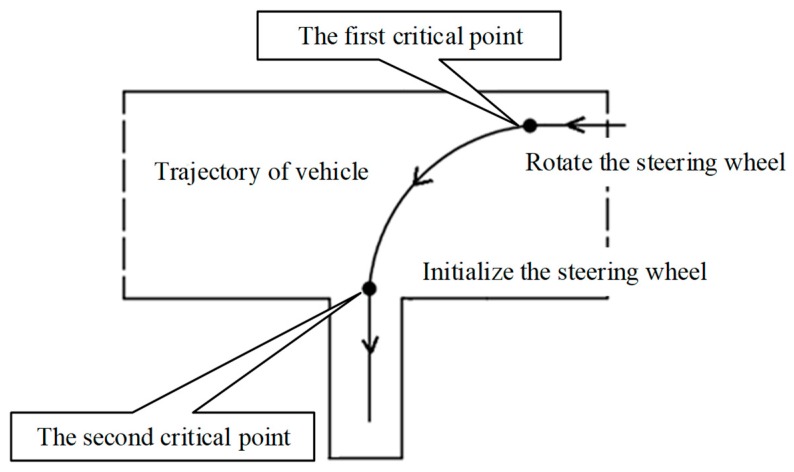
The process of reverse parking.

**Figure 11 sensors-19-00630-f011:**
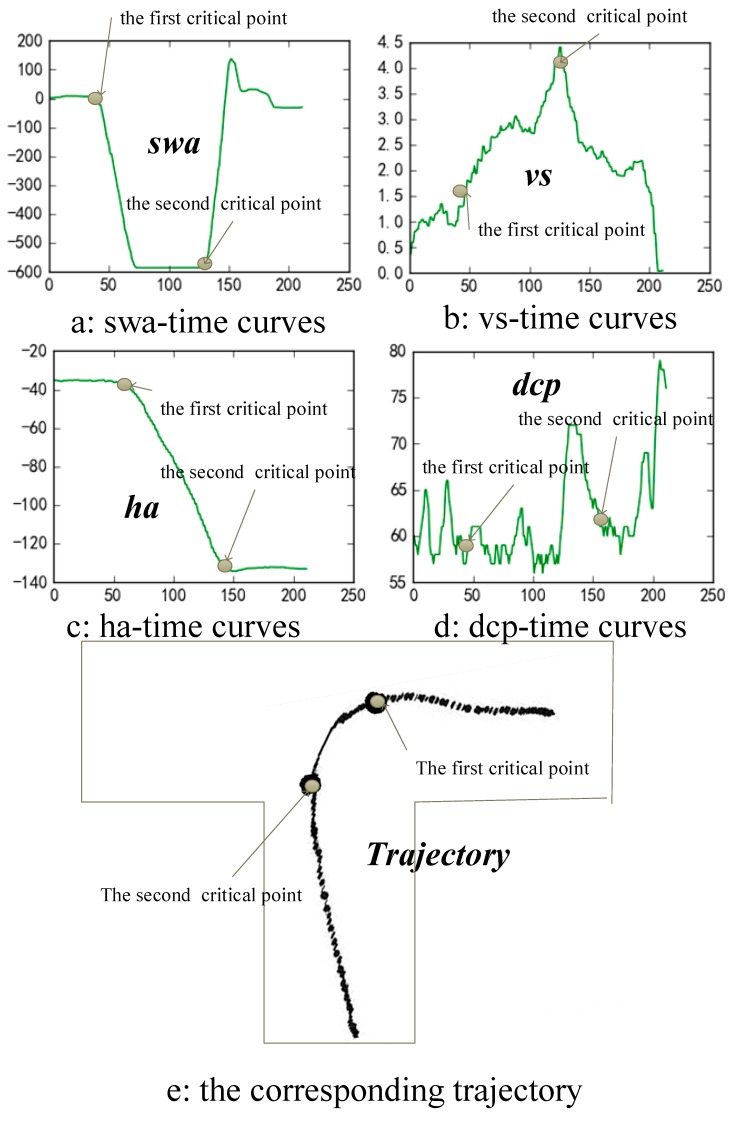
Curves of the four features in a specific reverse parking operation.

**Figure 12 sensors-19-00630-f012:**
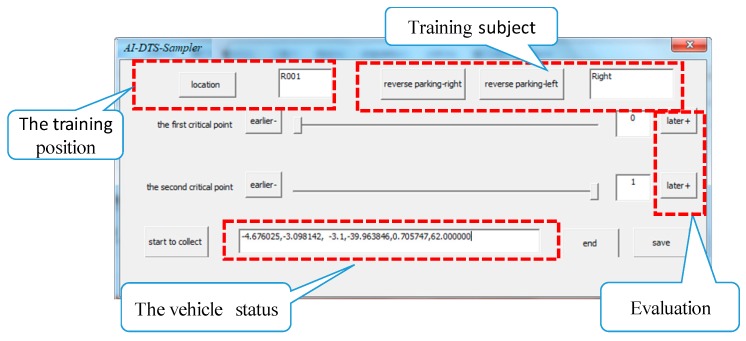
The AI-DTS-Sampler.

**Figure 13 sensors-19-00630-f013:**
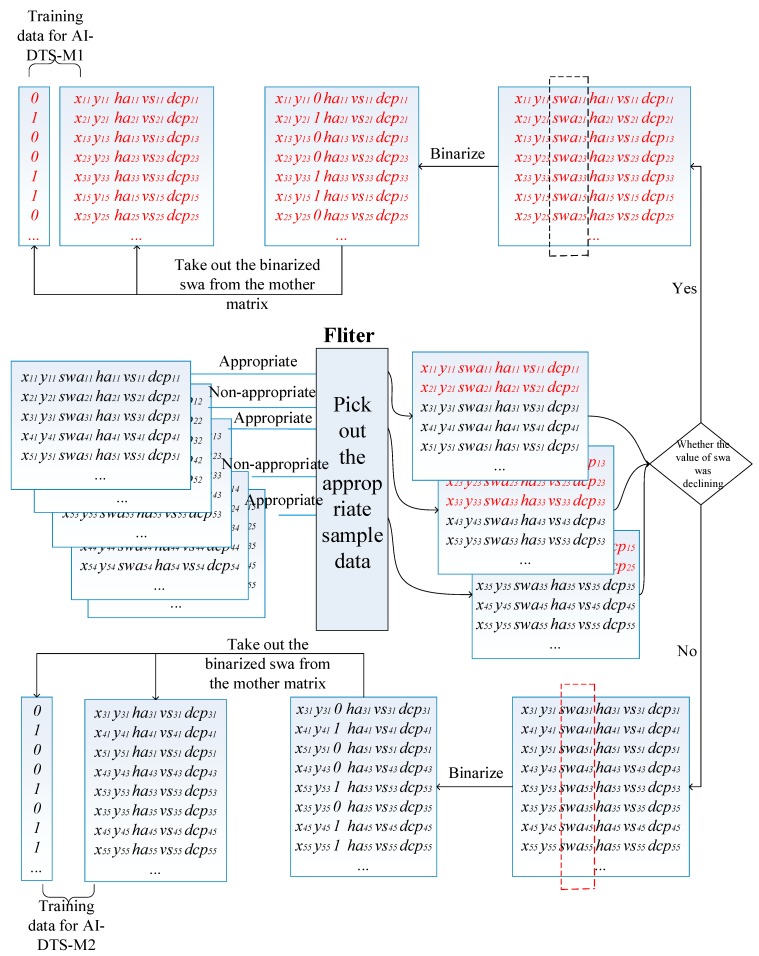
The process of sample data preprocessing.

**Figure 14 sensors-19-00630-f014:**
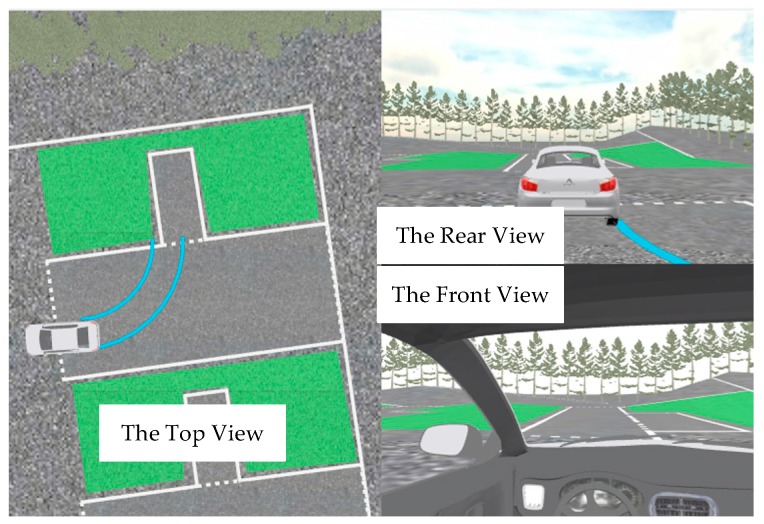
The interface of AI-DTS.

**Figure 15 sensors-19-00630-f015:**
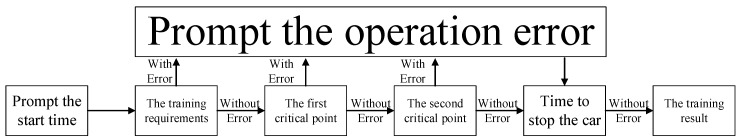
The voice prompt and their steps in the process of reverse parking.

**Figure 16 sensors-19-00630-f016:**
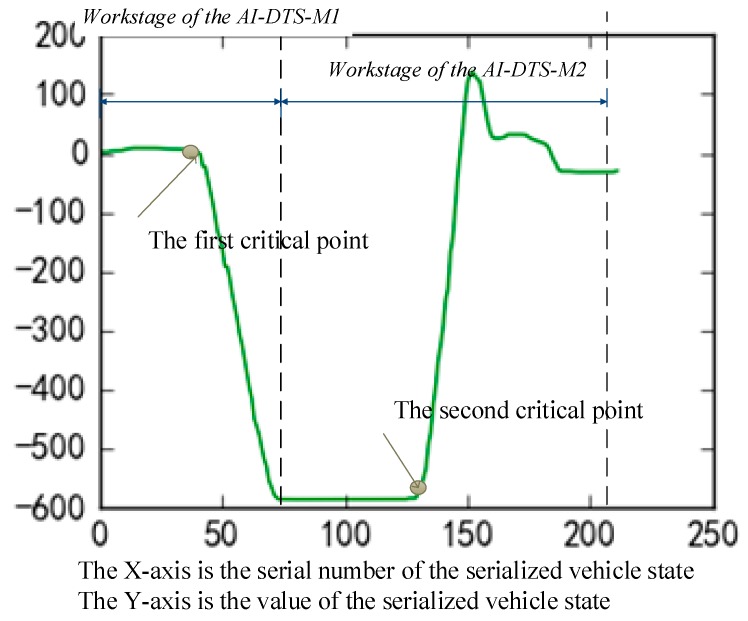
Work stage division of the AI-DTS-M1 and the AI-DTS-M2.

**Figure 17 sensors-19-00630-f017:**
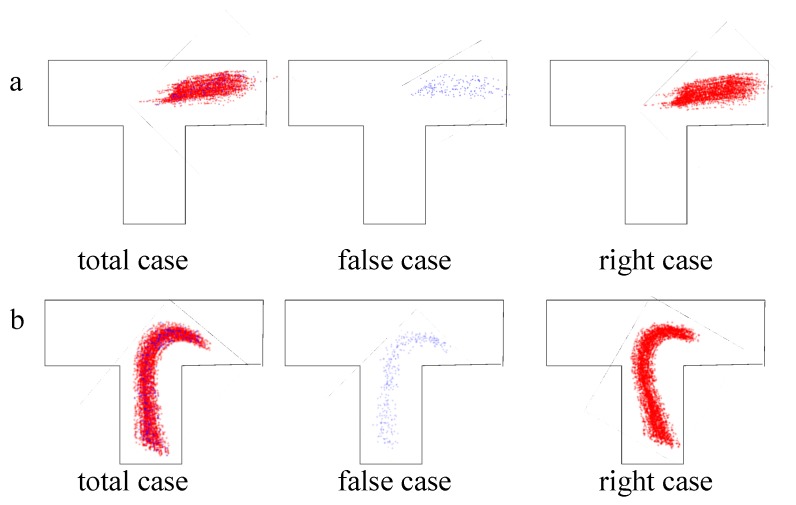
The distribution of the judgements on the two critical points: (**a**) The first critical point; and (**b**) The second critical point.

**Figure 18 sensors-19-00630-f018:**
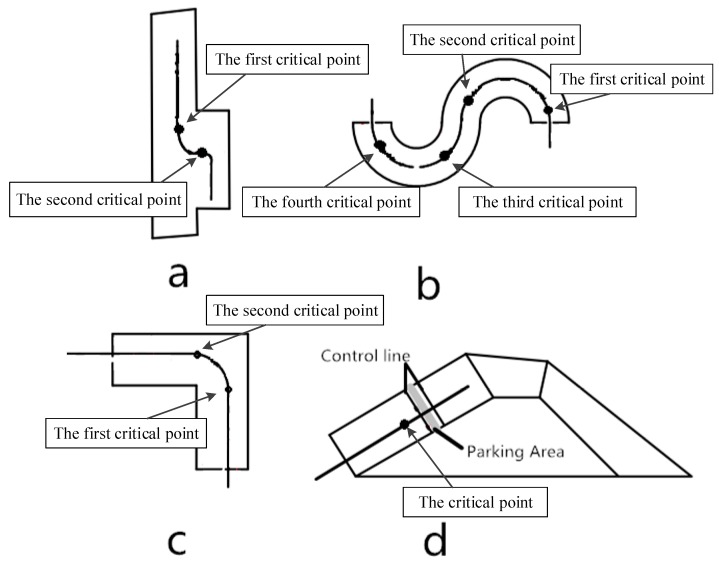
The critical points of the rotating steering wheel in the remaining four subjects (a) the critical point of Parallel parking; and (**b**) the critical point of Tortuous route; and (**c**) the critical point of the Quarter turn, and (**d**) the critical point of Ramp parking.

**Table 1 sensors-19-00630-t001:** The performance of the AI-DTS.

Model	Task	Task	Precision	Recall	Support
AI-DTS-M1	Predict the right moment of rotating the steering wheel in the first critical point	Rotate the steering wheel	0.95	0.93	2786
		Non-rotate the steering wheel	0.85	0.88	1255
		Total	0.92	0.92	4041
AI-DTS-M2	Predict the right moment of rotating the steering wheel in the second critical point	Rotate the steering wheel	0.94	0.94	3099
		Non-rotate the steering wheel	0.93	0.94	2717
		Total	0.94	0.94	5816

**Table 2 sensors-19-00630-t002:** Feedback of the experimenter.

Task	Whether the Judgment of AI-DTS and Trainee are Consistent		Whether the AI-DTS Suggest Rotating the Steering Wheel		Whether the Result is Acceptable	
	YES	NO	YES	NO	YES	NO
Predict the right moment of rotating the steering wheel in the first critical point	84	16	92	8	94	6
Predict the right moment of rotating the steering wheel in the second critical point	91	9	87	13		

**Table 3 sensors-19-00630-t003:** Improvement of AI-DTS in practical training.

	The Number of People Passing the Examination	The Number of People Who Failed to Pass the Examination.	The Pass Rate	The Average Training Time
Experimental group	84	16	84%	32
Reference group	62	38	62%	40
The improve	22	−22	22%	20%

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
