# Peer review of "An Intelligent Driver Training System Based on Real Cars"

_sensors, 2019, doi:10.3390/s19030630_

Reviewer 1 Report

The paper proposes a reverse parking machine learning model, core of an Artificial Intelligence-based Driver Training System (AI-DTS). This model, once trained, was developed to real-time coach driver during training and to provide post-assessment feedback. Model validity and reliability was tested comparing two samples of drivers trained using the developed method and the traditional one.

The paper is quite interesting and potentially useful to practitioners. However, the Driver Training System proposed is limited to the sole maneuver of “reverse parking”. The methodology should be described in a clearer and more exhaustive. Validity and reliability of model is not completely convincing. The discussion is lacking.

·         “Introduction” section: The motivation and scope of the study should be better and extensively specified. Please, indicate which gaps in the current literatures you want to address in your research, if any. I did not expect to find the results of the study (lines 76-79) in this section.

·         “The framework and the key technologies” section: The sentence in lines 83-84 needs one or more references.

·         “The framework and the key technologies” section: The characteristics of sample chosen for training the AI-DTS should be provided. The participants were drawn randomly ensuring a sample balanced respect the age, height and sex, is it true?  

·         “Performance assessment” section: Please provide also the characteristics of the two samples composed each of 100 volunteers (the first trained using the Intelligent Driving Training System, the latter via traditional training methods). How they were drawn? Do they have similar characteristics (socio-demographic, etc.)?

·         “Applicability assessment based on the experimenter's intuitive feedback” sub-section: There is a discrepancy between the text (lines 414) and the Table 2 (last row, third column): 91 vs 87.

·         “Applicability assessment based on the experimenter's intuitive feedback” sub-section: In the 6% of cases, the output of AI-DTS model can induce drivers to perform a wrong maneuver of “reverse parking”: the motivation and the implication on training of this not negligible percentage of false-positive should be adequately approached in the paper.

·         “AI-DTS's performance in training practice” sub-section, lines 422-423: No statistical analysis was performed so please remove the word “statistically” from the sentence.

·         “AI-DTS's performance in training practice” sub-section: Despite being the most difficult maneuver, the “reverse parking” isn’t the only driving maneuver on which it is based the “driver qualification examination”. For this reason, the interpretation of the results proposed by the authors is not convincing (unless all the drivers, traditionally trained, failed to pass the examination due to insufficient skills/training on the “reverse parking”). 

·         “AI-DTS's performance in training practice” sub-section: It is unclear how the “average training time” was assessed. Are you referring to the subjective judgment of the trainer? Are you referring only to the maneuver of “reverse parking” or to the whole training? If you consider the whole training time, what is the percentage reduction for a single maneuver of “reverse parking”?

·         A discussion should be added in the paper to address a general and critical evaluation of the proposed training method, stressing on its advantages and limitations. A comparison with the other methods in the literature is mandatory

·         The effective contribution to research literature should be more highlighted.

·         Figures 2 and 3: The quality of these figures should be enhanced. These figures are briefly mentioned in the text: they deserve more attention and space in the text.

·         Figure 5: The text in this figure is not readable.

·         Figure 7: “Head Angle” instead of “Head Angel”.

·         Table 2, last row first column: “Predict the right moment of rotating the steering wheel in the second critical point” instead of “Predict the right moment of rotating the steering wheel in the first critical point”

Author Response

Point 1: “Introduction” section: The motivation and scope of the study should be better and extensively specified. Please, indicate which gaps in the current literatures you want to address in your research, if any. I did not expect to find the results of the study (lines 76-79) in this section.

Response 1: The “Introduction” section has been revised, and more attention has been paid to the previous research in the field of driver training. The focus of this study is described. In the “Introduction” section, the description of the research results is removed.

Point 2: “The framework and the key technologies” section: The sentence in lines 83-84 needs one or more references.

Response 2: As reviewer suggested, we added references for the statement of “driving schools use the five typical subjects shown in the figure for driving training, according to which the authorities certify the students’ driving skills’”.

Point 3: “The framework and the key technologies” section: The characteristics of sample chosen for training the AI-DTS should be provided. The participants were drawn randomly ensuring a sample balanced respect the age, height and sex, is it true?

Response 3: In fact, we do have age, height and gender control in the process of recruiting volunteers at random. However, this is not the focus of this paper. We deleted descriptions of age, height and gender(lines 333-334).

Point 4: “Performance assessment” section: Please provide also the characteristics of the two samples composed each of 100 volunteers (the first trained using the Intelligent Driving Training System, the latter via traditional training methods). How they were drawn? Do they have similar characteristics (socio-demographic, etc.)?

Response 4: All the volunteers have no driving experience at all and it is mentioned in line 195 and line 458.

Point 5: “Applicability assessment based on the experimenter's intuitive feedback” sub-section: There is a discrepancy between the text (lines 414) and the Table 2 (last row, third column): 91 vs 87.

Response 5: We are very sorry for our negligence, we have corrected it now.

Point 6: “Applicability assessment based on the experimenter's intuitive feedback” sub-section: In the 6% of cases, the output of AI-DTS model can induce drivers to perform a wrong maneuver of “reverse parking”: the motivation and the implication on training of this not negligible percentage of false-positive should be adequately approached in the paper.

Response 6: In application, the system gives suggestions according to the state vector of 3-5 points. This greatly reduces the chance of giving incorrect advice. This is not the focus of this article, so we have not made a detailed introduction to it.

Point 7: “AI-DTS's performance in training practice” sub-section, lines 422-423: No statistical analysis was performed so please remove the word “statistically” from the sentence.

Response 7: It is really true as reviewer suggested that there is no statistical analysis was performed in this section. We have deleted it now.

Point 8: “AI-DTS's performance in training practice” sub-section: Despite being the most difficult maneuver, the “reverse parking” isn’t the only driving maneuver on which it is based the “driver qualification examination”. For this reason, the interpretation of the results proposed by the authors is not convincing (unless all the drivers, traditionally trained, failed to pass the examination due to insufficient skills/training on the “reverse parking”).

Response 8: In actually, the method of developing the AI-DTS which is introduced in this paper with the example of reverse parking is a general method. The AI-DTS includes all the training subjects. We have added an explanation to it in this paper(lines 507-512).

Point 9: “AI-DTS's performance in training practice” sub-section: It is unclear how the “average training time” was assessed. Are you referring to the subjective judgment of the trainer? Are you referring only to the maneuver of “reverse parking” or to the whole training? If you consider the whole training time, what is the percentage reduction for a single maneuver of “reverse parking”?

Response 9: The time mentioned in this section is the time required for a driver who has no driving experience at all to master driving skills to pass all training subjects. This study is focus people who have absolutely no driving experience, so we do not pay attention to the operating time of a single maneuver of "reverse parking".

Point 10: A discussion should be added in the paper to address a general and critical evaluation of the proposed training method, stressing on its advantages and limitations. A comparison with the other methods in the literature is mandatory.

Response 10: The discussion section was assed. In this section, we emphasize the universality of the methods introduced in this paper.

Point 11: The effective contribution to research literature should be more highlighted.

Response 11: The “Introduction” section has been revised, and more attention has been paid to the previous research in the field of driver training.

Point 12: Figures 2 and 3: The quality of these figures should be enhanced. These figures are briefly mentioned in the text: they deserve more attention and space in the text.

Response 12: A description of Figure 2 and Figure 3 has been added(lines 140-146).

Point 13: Figure 5: The text in this figure is not readable.

Response 13: The Figure 5 have been modified.

Point 14: Table 2, last row first column: “Predict the right moment of rotating the steering wheel in the second critical point” instead of “Predict the right moment of rotating the steering wheel in the first critical point”.

Response 14: The word “first” was replaced by “second”.

Special thanks to you for your good comments!

Reviewer 2 Report

Title should be modified so readers can get right away that the paper is on reverse parking maneuvers.

Authors should be careful in drawing conclusion solely based on the vehicle state since a car might go straight while the driver is asleep. Therefore, such a system should also include the driver's behaviors behind the wheel such as eye movements.

On line 60, authors refer to the operational control of the vehicle, based on Michon's model of driving, the training suggested here do lack in terms of tactical and strategic interventions that are often described as critical to safe driving since bad driving maneuvers are often preceded by poor decision making.

Line 72, 1000 groups, line 75, 200 groups ? is it individual driver or a group ? The N of participants has to be clarified. If these are groups, the total number of drivers should be added. Moreover, since we are applying here AI to detect proper maneuver, this section should state right away if there is real driver or not conducting the task or if these are automatically generated data.

In the model, it should be clarified that he parking is executed in a "one-shot" maneuver and does not allow the driver to go forward and back-up a second time if an initial error was made.

General comments and point for discussion

- what will be the impact of back-up camera on such a system since they are now mandatory in America since 2018 ?

- the use of DGPS systems makes it difficult to transfer to actual driving maneuvers since to allow for the precision required to provide such feedback, driver would have to operate in a really close area so the gps can work appropriately. This has to be consider in the limitations of the actual study.

Overall, it is an interesting paper on the of automation for detecting driving maneuver and allow for providing feedback to the driver. However, limitations should be brought forward since the system has described does not include the driver in the loop and how visual search is done. Therefore, I would suggest the authors to clarify this important issue to enhance the readability and applicability of their manuscript.

Author Response

Point 1: Title should be modified so readers can get right away that the paper is on reverse parking maneuvers.

Response 1: The method of developing the AI-DTS which is introduced in this paper with the example of reverse parking is a general method. The AI-DTS includes all the training subjects. We have added an explanation to it in this paper(lines 120-122, lines 507-512).

Point 2: Authors should be careful in drawing conclusion solely based on the vehicle state since a car might go straight while the driver is asleep. Therefore, such a system should also include the driver's behaviors behind the wheel such as eye movements.

Response 2: In China, for the safety of the trainees, there is a camera on each learner-driven vehicle to observe the movements of the trainees. Therefore, we did not pay attention to the eye movements of the trainee.

Point 3: On line 60, authors refer to the operational control of the vehicle, based on Michon's model of driving, the training suggested here do lack in terms of tactical and strategic interventions that are often described as critical to safe driving since bad driving maneuvers are often preceded by poor decision making.

Response 3: This is a driver training system. The safety is mainly ensured by the safety system on the learner-driven vehicle and the coach. We have deleted the topic of safe.

Point 4: Line 72, 1000 groups, line 75, 200 groups ? is it individual driver or a group ? The N of participants has to be clarified. If these are groups, the total number of drivers should be added. Moreover, since we are applying here AI to detect proper maneuver, this section should state right away if there is real driver or not conducting the task or if these are automatically generated data.

Response 4: A description of the number and group of volunteers is added(line192-194). The source of data is explained(line 194-195). The words “1000 groups of volunteers and coaches” was replaced by “1000 volunteers”. The words “200 groups of volunteer trainees” was replaced by “200 volunteer trainees”.

Point 5: In the model, it should be clarified that he parking is executed in a "one-shot" maneuver and does not allow the driver to go forward and back-up a second time if an initial error was made.

Response 5: It was clarified in line 77-78.

Point 6: what will be the impact of back-up camera on such a system since they are now mandatory in America since 2018 ?

Response 6: The AI-DTS needs to collect vehicle information including the angle of the steering wheel, the head angle, the vehicle speed and the depth of the clutch pedals. None of this has anything to do with back-up cameras. We believe that mandatory back-up cameras can dispel public doubts about intelligent driving systems.

Point 7: the use of DGPS systems makes it difficult to transfer to actual driving maneuvers since to allow for the precision required to provide such feedback, driver would have to operate in a really close area so the gps can work appropriately. This has to be consider in the limitations of the actual study.

Response 7: AI-DTS is aimed at drivers who have no driving experience at all(line 48-49). They usually train in driving schools(line 29-30).

Point 8: Overall, it is an interesting paper on the of automation for detecting driving maneuver and allow for providing feedback to the driver. However, limitations should be brought forward since the system has described does not include the driver in the loop and how visual search is done. Therefore, I would suggest the authors to clarify this important issue to enhance the readability and applicability of their manuscript.

Response 8: Thank you for your compliment. Although this study is inspired by autopilot, it focuses on helping people who have no driving experience to complete the exam subjects. Their training uses very limited and well-defined areas.

Special thanks to you for your good comments!

Reviewer 3 Report

Thank you for the opportunity to review manuscript entitled ‘An intelligent driver training system based on real car’. The article is presenting a machine learning approach to evaluate how learner drivers conduct a reverse parking. It was also used within an HMI to help train learners and improve their skills before their driving test. While this topic is of interest, there are a number of issues that would need to be addressed before the paper is publishable:

-          The title does not reflect the actual content of the paper: the authors did not develop a driver training system. They rather developed a reverse parking training system.

-          The literature review is very limited, and includes 14 of the 16 references in one paragraph. The review is quite long on broad teaching examples without discussing their paradigm, and fails to be very specific on driver training. In particular, authors mention that no research has been done ‘on real cars’, which is not an accurate statement. I refer them to the following paper for instance: Malik, Husnain, Larue, Gregoire S., Rakotonirainy, Andry, & Maire, Frederic D. (2014) Fuzzy logic to evaluate driving maneuvers: An integrated approach to improve training. IEEE Transactions on Intelligent Transportation Systems, 16(4), pp. 1728-1735.

-          The main issue with evaluating the value of the article is the lack of specific details on the methodology, data analysis and results. While the paper is quite long, a lot of the text is repetitive and uninformative. The paper tries to deliver a lot: finding the best factors for parking, training an algorithm, testing the algorithm, creating an HMI, training learner drivers and testing its effects. However, it fails to report anything in details. For instance, there are a lot of general equations without defined variables, there is no presentation of how the information is provided to learners for the HMI, etc.

-          There is also a lack of justification on why this approach would be necessary, better than any other approaches.

-          Most figures are barely legible.

-          English should be proofread.

Author Response

Point 1: The title does not reflect the actual content of the paper: the authors did not develop a driver training system. They rather developed a reverse parking training system. 

Response 1: The method of developing the AI-DTS which is introduced in this paper with the example of reverse parking is a general method. The AI-DTS includes all the training subjects. We have added an explanation to it in this paper(lines 120-122, lines 507-512).

Point 2: The literature review is very limited, and includes 14 of the 16 references in one paragraph. The review is quite long on broad teaching examples without discussing their paradigm, and fails to be very specific on driver training. In particular, authors mention that no research has been done ‘on real cars’, which is not an accurate statement. I refer them to the following paper for instance: Malik, Husnain, Larue, Gregoire S., Rakotonirainy, Andry, & Maire, Frederic D. (2014) Fuzzy logic to evaluate driving maneuvers: An integrated approach to improve training. IEEE Transactions on Intelligent Transportation Systems, 16(4), pp. 1728-1735.

Response 2: The “Introduction” section has been revised, and more attention has been paid to the previous research in the field of driver training. The focus of this study is described.

Point 3: The main issue with evaluating the value of the article is the lack of specific details on the methodology, data analysis and results. While the paper is quite long, a lot of the text is repetitive and uninformative. The paper tries to deliver a lot: finding the best factors for parking, training an algorithm, testing the algorithm, creating an HMI, training learner drivers and testing its effects. However, it fails to report anything in details. For instance, there are a lot of general equations without defined variables, there is no presentation of how the information is provided to learners for the HMI, etc.

Response 3: Some general equations are deleted since that they are not the core of this study. Human-computer interaction is introduced in the section 2.3.4.

Point 4: There is also a lack of justification on why this approach would be necessary, better than any other approaches.

Response 4: It was described in the revised ‘introduction’ section. This study is focus on helping trainer without any driving experience to pass subject prescribed in the driving skills identification test in China.

Point 5: Most figures are barely legible.

Response 5: The Figure 1 and Figure 5 have been modified and a description of Figure 2 and Figure 3 has been added.

Point 6: English should be proofread.

Response 6: We have checked the expression, grammar, and typos throughout the paper.

Special thanks to you for your good comments!

Round  2

Reviewer 1 Report

My comments were adequately addressed with a substantial improving of the paper. However, I consider useful to add socio-demographic characteristics (at least age, sex and height) of the different samples used in the study, both to improve the comprehensibility of the paper and to strengthen the results of the analysis on the learning efficiency of the AI-DTS.

·         Line 234: Please, add a space between the words “assessment” and “parameter”.

·         “Acknowledgments”, lines 556-558: Considering that no acknowledgment was made by authors, this section should be deleted.

Author Response

Response to Reviewer 1 Comments

Point 1: My comments were adequately addressed with a substantial improving of the paper. However, I consider useful to add socio-demographic characteristics (at least age, sex and height) of the different samples used in the study, both to improve the comprehensibility of the paper and to strengthen the results of the analysis on the learning efficiency of the AI-DTS. 

Response 1: We are pleased to note the favorable comments of the  opening sentence. As reviewer suggested, we checked the information of volunteers and introduced the age, gender and height distribution information of volunteers in this paper.(lines 192-194, Figure 6 and Figure 7)

Point 2: Line 234: Please, add a space between the words “assessment” and “parameter”.

Response 2: Thanks for your conscientiousness, a space has been added between the words “assessment” and “parameter”(line 273).

Point 3: “Acknowledgments”, lines 556-558: Considering that no acknowledgment was made by authors, this section should be deleted.

Response 3: The section “Acknowledgments” and the section “Conflicts of Interest” have been deleted (lines 603-606).

Thanks very much for your kind work and consideration on our paper. The Chinese New Year is coming soon. Happy New Year to you!

Reviewer 2 Report

Dear authors

Thanks for your responses and the modifications made to your manuscript.

Regards,

-Reviewer1

Author Response

Response to Reviewer 2 Comments

Thanks very much for your kind work and consideration on our paper. The Chinese New Year is coming soon. Happy New Year to you!
